# MetaStreet: Semi-Supervised Multimodal Learning for Street-Level Socioeconomic Prediction

**Meng Chen** [1]  **Junjie Yang** [1]  **Zechen Li** [1 2]  **Kai Zhao** [3]  **Hongjun Dai** [1]  **Weiming Huang** [4]

## Abstract

Predicting street-level socioeconomic indicators from street view imagery is fundamental to urban planning. Existing methods typically extract visual features via pretrained encoders and propagate information through graph-based learning, but they fail to fully exploit the structured, task-relevant, and label-efficient learning signals inherent in urban scenes. We propose MetaStreet, a semi-supervised multimodal framework with three components: (1) a semantic-spatial visual encoder that jointly models object co-occurrence and spatial adjacency at the semantic category level, (2) a task-aware textual encoder that steers LLMs toward prediction-relevant features via task-specific prompts, and (3) a geography-aware graph contrastive learning module that leverages spatial autocorrelation to extend contrastive supervision to unlabeled streets, enabling them to actively participate in representation learning. Experiments on two cities across three socioeconomic prediction tasks demonstrate that MetaStreet consistently outperforms state-of-the-art methods.

## 1. Introduction

Street-level urban sensing, which predicts socioeconomic indicators such as land-use function, economic activity, and property values from street view imagery, is fundamental to urban planning, resource allocation, and policy-making (Xia et al., 2021; Zhou et al., 2024; Chen et al., 2025a). The proliferation of street view imagery (SVI), with its extensive geographic coverage and rich human-perspective visual semantics, has opened new opportunities for automated, scalable urban analysis (Zhang et al., 2024; Huang et al., 2024). Existing methods typically leverage pretrained encoders to extract feature representations from street view images, and then employ graph-based learning to propagate information across the street network for socioeconomic prediction (Chen et al., 2024; Zhang et al., 2023). However, these approaches share a common limitation: they fail to fully exploit the *structured*, *task-relevant*, and *label-efficient* learning signals inherent in urban scenes. We identify three interrelated challenges corresponding to these aspects.

**C1: Inadequate modeling of compositional structure in visual encoding.** Urban scenes are inherently compositional: the combination of semantic objects, including both their co-occurrence patterns and spatial arrangements, conveys critical socioeconomic information. As illustrated in Figure 1, residential and commercial streets may share similar object proportions yet differ markedly in spatial organization. Existing approaches fail to capture this compositional structure effectively. Pretrained encoders (Ouyang et al., 2024; Chen et al., 2025b) learn domain-agnostic features without distinguishing socioeconomically relevant elements, and lack explicit modeling of relationships between semantic categories. Segmentation-based methods (Fan et al., 2023) operate at the semantic level but discard spatial relationships by treating categories independently. Neither approach jointly models semantic co-occurrence and spatial adjacency.

**C2: Task-agnostic textual knowledge extraction.** Recent advances leverage multimodal large language models to generate textual descriptions from street view images (Schumann et al., 2024; Li et al., 2024a). However, these descriptions are task-agnostic: a caption emphasizing "vintage brick architecture" benefits neighborhood age prediction but provides little signal for economic activity prediction, which requires attention to commercial signage and pedestrian density. This mismatch between generic captions and specific prediction objectives limits the utility of textual features for targeted socioeconomic inference.

**C3: Passive treatment of unlabeled data.** Acquiring street-level socioeconomic labels is labor-intensive, resulting in extreme label scarcity. While semi-supervised graph meth-

[1]School of Software, Shandong University, Jinan, China [2]Key Laboratory of Urban Land Resources Monitoring and Simulation, Ministry of Natural Resources, Shenzhen, China [3]Workday AI, Pleasanton, USA [4]School of Geography, University of Leeds, Leeds, UK. Correspondence to: Hongjun Dai <dahogn@sdu.edu.cn>.

*Proceedings of the $43^{rd}$ International Conference on Machine Learning*, Seoul, South Korea. PMLR 306, 2026. Copyright 2026 by the author(s).

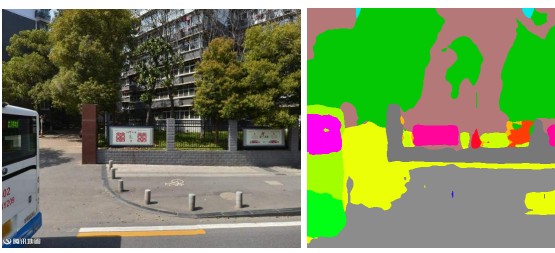

*(a)* Residential Area

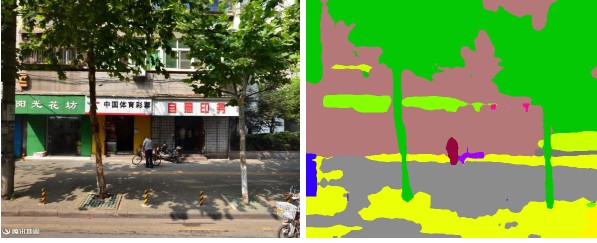

*(b)* Commercial Area

*Figure 1.* Two street scenes with similar object proportions but distinct spatial arrangements. In the residential area (a), vegetation separates buildings from roads. In the commercial area (b), buildings directly face the street. These spatial patterns provide discriminative cues that proportion-based methods cannot capture.

ods (Chen et al., 2024; Zhang et al., 2023) propagate labels through spatial adjacency, they treat unlabeled nodes as passive recipients of information flow. The representation learning objective operates only on the limited labeled set, leaving the vast majority of streets excluded from the core learning process.

These challenges reflect a natural progression in the field: early visual encoders left scene structure unmodeled, subsequent multimodal methods introduced textual signals but in a task-agnostic manner, and the latest graph-based approaches incorporated spatial context yet kept unlabeled nodes passive. Under extreme label scarcity, these unresolved challenges become tightly coupled: inadequate visual and textual representations limit the discriminative power of embeddings learned from labeled streets, while the failure to actively engage unlabeled data exacerbates the impact of label scarcity. This motivates a unified framework that jointly extracts richer multimodal representations and expands the effective training signal beyond the labeled set.

We propose **MetaStreet**, a semi-supervised multimodal framework that addresses the above challenges through three targeted components. To tackle **C1**, we design a *semantic-spatial visual encoder* that jointly models object co-occurrence and spatial adjacency at the semantic category level, capturing the compositional structure that pretrained encoders and proportion-based methods both miss. To address **C2**, we introduce the *task-aware textual encoder* that aggregates image-level captions into street-level documents and prepends task-specific prompts, steering LLMs toward

prediction-relevant features. To resolve **C3**, we develop *geography-aware graph contrastive learning* that leverages spatial autocorrelation to extend contrastive supervision to unlabeled streets, enabling them to actively participate in representation learning.

Our main contributions are summarized as follows:

- We propose MetaStreet, a multimodal semi-supervised framework for street-level socioeconomic prediction. To our knowledge, this is the first work that jointly addresses compositional visual encoding, task-aware textual extraction, and label-efficient learning in a unified framework, establishing a new paradigm for street-level urban understanding.

- We introduce a semantic-spatial visual encoding paradigm for urban scene understanding. Unlike pretrained encoders that produce generic features and segmentation methods that discard spatial relationships, our approach operates at the semantic category level while preserving compositional structure. This paradigm offers the urban computing community a new perspective on encoding street-level imagery when scene structure is the key discriminative signal.

- We propose geography-aware graph contrastive learning that activates unlabeled nodes in the representation learning process. This mechanism demonstrates how domain-specific priors, such as spatial autocorrelation, can be leveraged to extend supervision beyond labeled data, providing a transferable strategy for label-scarce scenarios in urban analytics and related domains.

- Extensive experiments across two cities and three socioeconomic prediction tasks demonstrate consistent improvements over state-of-the-art methods. We release our code and processed datasets at https://github.com/AIMUrban/MetaStreet.

## 2. Problem Formulation

**Definition 2.1 (Urban Street Network).** We model the urban area as a street network $\mathcal{G}_s = (\mathcal{S}, \mathcal{E})$, where $\mathcal{S} = \{s_1, s_2, \ldots, s_n\}$ denotes the set of $n$ street segments extracted from OpenStreetMap[1], and $\mathcal{E} \subseteq \mathcal{S} \times \mathcal{S}$ represents spatial proximity relationships between streets. Specifically, an edge $(s_i, s_j) \in \mathcal{E}$ exists if street $s_j$ is among the $k$-nearest neighbors of $s_i$ based on geographic distance.

**Definition 2.2 (Street View Image Set).** For each street $s_i \in \mathcal{S}$, we collect a set of street view images $\mathcal{I}_i = \{I_{i,1}, I_{i,2}, \ldots, I_{i,m_i}\}$, where $m_i = |\mathcal{I}_i|$ denotes the number of images associated with street $s_i$. These images are

---

[1]https://www.openstreetmap.org

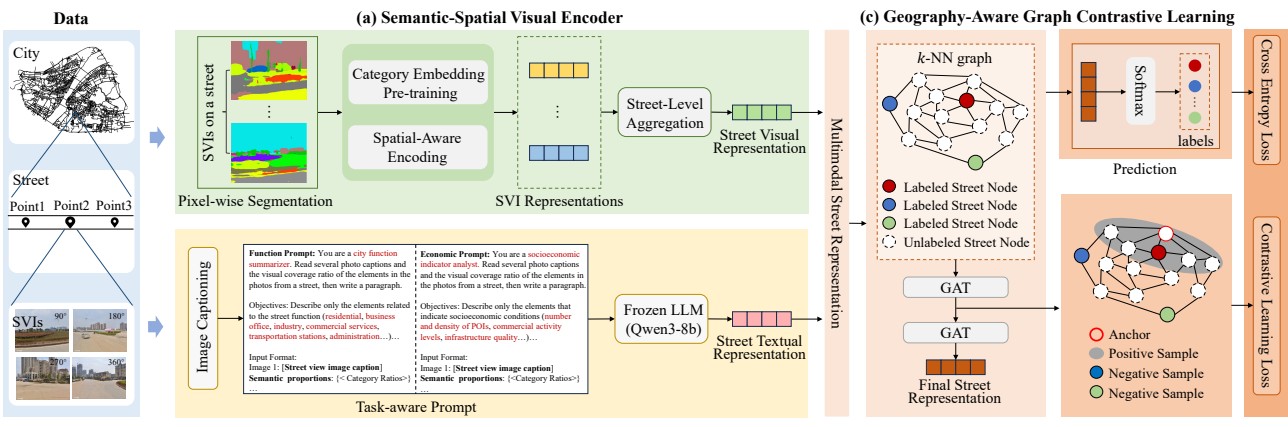

*Figure 2.* The overall architecture of MetaStreet. (a) The semantic-spatial visual encoder models object co-occurrence and spatial adjacency to capture compositional structure (addressing **C1**). (b) The task-aware textual encoder aggregates image captions with task-specific prompts to extract prediction-relevant features (addressing **C2**). (c) Geography-aware graph contrastive learning extends contrastive supervision to unlabeled streets via spatial proximity (addressing **C3**).

captured at discrete sampling points along the street from multiple viewpoints, providing comprehensive visual coverage of the urban built environment. Notably, $m_i$ varies across streets due to differences in street length and sampling density.

**Definition 2.3** (**Semi-Supervised Socioeconomic Prediction**). Let $\mathcal{Y} = \{1, 2, \ldots, C\}$ denote the set of $C$ class labels for a socioeconomic indicator (e.g., street function, economic activity level, or house price tier). In the semi-supervised setting, only a small subset of streets $\mathcal{S}^L = \{s_1, s_2, \ldots, s_l\} \subset \mathcal{S}$ are associated with ground-truth labels $\{y_1, y_2, \ldots, y_l\}$, while the remaining streets $\mathcal{S}^U = \mathcal{S} \setminus \mathcal{S}^L$ remain unlabeled. The objective is to predict the socioeconomic label $y_i \in \mathcal{Y}$ for each street $s_i \in \mathcal{S}$, leveraging both its street view images $\mathcal{I}_i$ and the spatial relationships encoded in the street network $\mathcal{G}_s$.

## 3. Methodology

We present MetaStreet, a semi-supervised multimodal learning framework for street-level socioeconomic prediction. As illustrated in Figure 2, MetaStreet comprises three core components: (1) a semantic-spatial visual encoder that models both object co-occurrence and spatial adjacency from segmented imagery; (2) a task-aware textual encoder that guides LLMs toward prediction-relevant cues via task-specific prompts; and (3) a geography-aware graph contrastive learning module that enables unlabeled streets to actively participate in representation learning. Finally, we describe the joint training objective.

### 3.1. Semantic-Spatial Visual Encoder

We propose a semantic-spatial visual encoder that models (i) semantic co-occurrence, capturing which object categories

tend to appear together, and (ii) spatial adjacency, capturing how objects are arranged within each image.

#### 3.1.1. PIXEL-WISE SEMANTIC SEGMENTATION

Given a street view image $I_{i,j}$ (the $j$-th image of street $s_i$), we apply a pretrained semantic segmentation model (Zhou et al., 2019) to obtain a pixel-wise label map $\mathbf{M}_{i,j} \in \{1, \ldots, K\}^{H \times W}$, where $K$ denotes the number of object categories and $H \times W$ is the image resolution.

We derive a semantic proportion vector $\mathbf{p}_{i,j} \in \mathbb{R}^K$, where the $k$-th entry represents the normalized proportion of pixels belonging to category $k$:

$$\mathbf{p}_{i,j}[k] = \frac{|\{(h, w) : \mathbf{M}_{i,j}[h, w] = k\}|}{H \times W}. \quad (1)$$

While this vector provides a compact summary of scene composition, it treats each category independently and ignores both semantic relationships and spatial arrangements.

#### 3.1.2. CATEGORY EMBEDDING PRETRAINING

To capture semantic relationships between object categories, we learn category embeddings from their co-occurrence patterns across the image corpus. The key insight is that categories frequently appearing together in urban scenes share semantic affinity.

We treat each image as a "document" and the object categories present in it as "words". We apply the Skip-gram model (Mikolov et al., 2013) to learn category embeddings by maximizing the likelihood of observing co-occurring categories:

$$\max_{\mathbf{C}} \sum_{I_{i,j}} \sum_{c_k \in \mathcal{C}_{i,j}} \sum_{c_l \in \mathcal{C}_{i,j} \setminus \{c_k\}} \log P(c_l | c_k; \mathbf{C}), \quad (2)$$

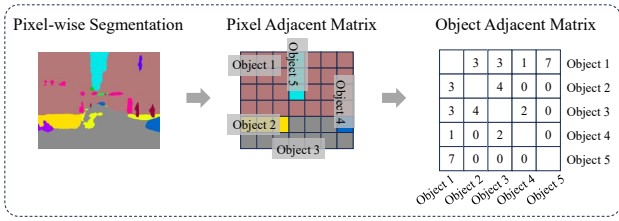

*Figure 3.* Computing the object adjacency matrix by counting pixel-level adjacencies between category pairs using 4-connected neighborhoods.

where $\mathcal{C}_{i,j}$ denotes the set of categories present in image $I_{i,j}$, and $\mathbf{C} \in \mathbb{R}^{K \times d_c}$ is the learned category embedding matrix. Each row $\mathbf{c}_k \in \mathbb{R}^{d_c}$ encodes the $k$-th category in a continuous semantic space where semantically related categories are closer.

### 3.1.3. SPATIAL-AWARE ENCODING

Beyond semantic composition, the spatial arrangement of objects provides crucial structural information. We construct an object adjacency matrix that captures pairwise spatial relationships between segmented categories.

For each image $I_{i,j}$, we compute the adjacency matrix $\mathbf{A}_{i,j} \in \mathbb{R}^{K \times K}$ by counting pixel-level adjacencies between category pairs in the segmentation map (see Figure 3):

$$\mathbf{A}_{i,j}[k,l] = \sum_{(h,w)} \sum_{(h',w') \in \mathcal{N}_4(h,w)} \mathbb{I}\big[\mathbf{M}_{i,j}[h,w] = k \\ \wedge \mathbf{M}_{i,j}[h',w'] = l\big], \quad (3)$$

where $\mathcal{N}_4(h,w)$ denotes the 4-connected neighborhood of pixel $(h,w)$, and $\mathbb{I}[\cdot]$ is the indicator function. Intuitively, $\mathbf{A}_{i,j}[k,l]$ measures how frequently categories $k$ and $l$ are spatially adjacent.

To retain category self-information during aggregation, we augment the adjacency matrix with an identity term:

$$\tilde{\mathbf{A}}_{i,j} = \mathbf{A}_{i,j} + \mathbf{I}_K, \quad (4)$$

where $\mathbf{I}_K$ is the $K \times K$ identity matrix.

We compute the spatial-aware image representation by integrating the adjacency structure, semantic proportions, and category embeddings:

$$\mathbf{S}_{i,j} = \tilde{\mathbf{A}}_{i,j} \cdot \text{diag}(\mathbf{p}_{i,j}) \cdot \mathbf{C}, \quad (5)$$

where $\text{diag}(\mathbf{p}_{i,j}) \in \mathbb{R}^{K \times K}$ converts the proportion vector into a diagonal scaling matrix. The resulting $\mathbf{S}_{i,j} \in \mathbb{R}^{K \times d_c}$ encodes each category's embedding weighted by its prevalence and aggregated with spatially adjacent categories. We flatten $\mathbf{S}_{i,j}$ into $\mathbf{v}_{i,j} = \text{flatten}(\mathbf{S}_{i,j}) \in \mathbb{R}^{K \cdot d_c}$.

### 3.1.4. STREET-LEVEL AGGREGATION

Each street contains multiple images captured sequentially along its length. Since images sampled at nearby locations tend to exhibit similar visual patterns, we employ an LSTM (Hochreiter & Schmidhuber, 1997) to model this spatial dependency when aggregating per-image representations:

$$\mathbf{e}_i^{\text{vis}} = \text{LSTM}(\mathbf{v}_{i,1}, \mathbf{v}_{i,2}, \ldots, \mathbf{v}_{i,m_i}), \quad (6)$$

where $m_i = |\mathcal{I}_i|$ is the number of images for street $s_i$, and $\mathbf{e}_i^{\text{vis}} \in \mathbb{R}^d$ is the final visual embedding.

## 3.2. Task-Aware Textual Encoder

Textual descriptions of street view images provide complementary high-level semantic information (Koh et al., 2023). We propose a task-aware textual encoder that (i) aggregates image-level descriptions at the street level, and (ii) uses task-specific prompts to guide LLMs toward extracting task-relevant features.

### 3.2.1. STREET-LEVEL CAPTION AGGREGATION

For each street view image $I_{i,j}$, we generate a descriptive caption $d_{i,j}$ using a pretrained multimodal model (Lin et al., 2024). Additionally, we convert the semantic segmentation statistics into a structured textual description $t_{i,j}$ (e.g., "`building: 30%, vegetation: 25%, road: 20%, ...`"). For street $s_i$, we aggregate all image-level descriptions into a unified street document:

$$D_i^{\text{raw}} = \bigoplus_{j=1}^{m_i} (d_{i,j} \oplus t_{i,j}), \quad (7)$$

where $\oplus$ denotes string concatenation.

### 3.2.2. TASK-SPECIFIC PROMPT DESIGN

Different socioeconomic prediction tasks require attention to different aspects of street scenes. To guide the LLM toward task-relevant information, we prepend a task-specific prompt $p_{\text{task}}$ to the street document:

$$D_i = p_{\text{task}} \oplus D_i^{\text{raw}}. \quad (8)$$

For the street function prediction task, the prompt may be: "Based on the following street scene descriptions, identify visual cues that indicate the primary function of this street. Focus on building types, land use patterns, and activity indicators." The complete prompt templates for all prediction tasks are provided in Appendix A.

The complete document $D_i$ is processed by a pretrained LLM (Qwen3-8B (Yang et al., 2025)) with frozen weights. We extract the final hidden state and project it through an MLP:

$$\mathbf{e}_i^{\text{txt}} = \text{MLP}(\text{LLM}(D_i)) \in \mathbb{R}^d. \quad (9)$$

To combine the visual and textual modalities, we employ a cross-attention fusion mechanism that computes adaptive weights for each modality based on the input, yielding the fused multimodal representation $\mathbf{e}_i \in \mathbb{R}^d$ for street $s_i$.

### 3.3. Geography-Aware Graph Contrastive Learning

With limited labeled streets, learning from individual labeled samples alone is insufficient. We introduce graph contrastive learning with geography-aware negative sampling to fully leverage unlabeled streets.

#### 3.3.1. GRAPH-BASED FEATURE PROPAGATION

Based on the spatial proximity graph $\mathcal{G}_s$ defined in Section 2, we employ a Graph Attention Network (GAT) (Veličković et al., 2018) to propagate and refine street representations:

$$\mathbf{z}_i = \text{GAT}(\mathbf{e}_i, \{\mathbf{e}_j : s_j \in \mathcal{N}_k(s_i)\}), \qquad (10)$$

where $\mathbf{z}_i \in \mathbb{R}^d$ is the graph-enhanced representation of street $s_i$, and $\mathcal{N}_k(s_i)$ denotes the $k$-nearest neighbors of $s_i$.

The final prediction is obtained via a softmax classifier:

$$\hat{\mathbf{y}}_i = \text{softmax}(\mathbf{W}_c \mathbf{z}_i + \mathbf{b}_c), \qquad (11)$$

where $\hat{\mathbf{y}}_i \in \mathbb{R}^C$ is the predicted probability distribution over $C$ classes.

For labeled streets $\mathcal{S}^L$, we define the supervised cross-entropy loss:

$$\mathcal{L}_{\text{sup}} = -\frac{1}{|\mathcal{S}^L|} \sum_{s_i \in \mathcal{S}^L} \sum_{c=1}^{C} y_{ic} \log \hat{y}_{ic}, \qquad (12)$$

where $y_{ic} \in \{0, 1\}$ is the ground-truth indicator for class $c$.

#### 3.3.2. CONTRASTIVE LEARNING WITH GEOGRAPHY-AWARE NEGATIVE SAMPLING

While the supervised loss trains the model on labeled streets, the abundant unlabeled streets remain underutilized. To actively leverage unlabeled data, we introduce a graph contrastive learning objective.

For positive sample construction, given each anchor street $s_i$, we define its positive sample as the mean representation of its 1-hop spatial neighbors:

$$\mathbf{z}_i^+ = \frac{1}{|\mathcal{N}_k(s_i)|} \sum_{s_j \in \mathcal{N}_k(s_i)} \mathbf{z}_j^{(1)}, \qquad (13)$$

where $\mathbf{z}_j^{(1)}$ denotes the embedding from the first GAT layer.

A key challenge is constructing meaningful negative samples when labels are scarce. Conventional approaches that rely solely on labeled samples for negative selection

severely limit the number of effective anchors. We propose a *geography-aware negative sampling strategy* that enables unlabeled streets to participate in contrastive learning. For labeled anchors, negative samples are other labeled streets with different class labels. For unlabeled anchors, we inherit negative samples from their labeled neighbors: if an unlabeled street $s_i$ is connected to labeled neighbor $s_j$ with label $y_j$, then labeled streets with labels different from $y_j$ serve as negative candidates for $s_i$. This inheritance is grounded in spatial autocorrelation, where geographically close streets tend to share similar socioeconomic characteristics. We empirically verify this assumption in Appendix G.

During training, we randomly sample $N_{\text{neg}}$ instances from each anchor's negative candidate set to form the final negative set $\mathcal{N}_i^-$. Following the InfoNCE framework (Oord et al., 2018), the contrastive loss is computed as:

$$\mathcal{L}_{\text{con}} = -\frac{1}{|\mathcal{S}_{\text{anchor}}|} \sum_{s_i \in \mathcal{S}_{\text{anchor}}} \log \frac{\exp(\text{sim}(\mathbf{z}_i, \mathbf{z}_i^+)/\tau)}{Z_i},$$

$$Z_i = \exp(\frac{\text{sim}(\mathbf{z}_i, \mathbf{z}_i^+)}{\tau}) + \sum_{\mathbf{z}_j^- \in \mathcal{N}_i^-} \exp(\frac{\text{sim}(\mathbf{z}_i, \mathbf{z}_j^-)}{\tau}). \qquad (14)$$

where $\mathbf{z}_i$ and $\mathbf{z}_j^-$ are obtained from the first GAT layer, $\text{sim}(\cdot, \cdot)$ denotes cosine similarity, $\tau$ is a temperature hyperparameter, and $\mathcal{S}_{\text{anchor}}$ includes both labeled and unlabeled streets that have valid negative candidates.

### 3.4. Joint Training Objective

The overall training objective combines supervised and contrastive losses:

$$\mathcal{L} = \lambda \mathcal{L}_{\text{sup}} + (1 - \lambda)\mathcal{L}_{\text{con}}, \qquad (15)$$

where $\lambda \in [0, 1]$ balances the two terms. The supervised loss ensures accurate predictions on labeled streets, while the contrastive loss encourages discriminative representations by leveraging spatial structure across all streets.

## 4. Experiments

### 4.1. Experimental Setup

#### 4.1.1. DATASETS

We evaluate MetaStreet on real-world urban datasets from two major Chinese cities: Wuhan and Xi'an (Zhang et al., 2023; Chen et al., 2024). The Wuhan dataset consists of 5,458 street segments, with 57,396 street view images collected from Tencent Maps. The Xi'an dataset comprises 4,918 segments with 100,956 images from Baidu Maps.

#### 4.1.2. PREDICTION TASKS

We design three socioeconomic prediction tasks to comprehensively evaluate street representation quality.

*Table 1.* Performance comparison on the Wuhan dataset, with the best results in bold and the second-best underlined.

| Category | Method | Street Function Prediction | | | | Economic Activity Prediction | | |
|---|---|---|---|---|---|---|---|---|
| | | ACC@1 | ACC@3 | Weighted-F1 | MRR | ACC | Macro-F1 | MRR |
| Visual | SemSeg | 29.87 | 56.79 | 30.81 | 49.21 | 36.74 | 35.53 | 60.57 |
| Visual | ResNet | 36.46 | 66.43 | 36.93 | 55.62 | 37.92 | 35.58 | 61.49 |
| Visual | READ | 37.86 | 65.19 | 33.02 | 56.02 | 29.05 | 21.44 | 55.11 |
| VLM | Qwen3-VL-235B | 35.92 | 68.38 | 34.53 | 54.88 | 34.67 | 32.65 | 60.96 |
| VLM | Qwen3-VL-8B (FT) | 37.54 | 72.69 | 39.38 | 57.68 | 38.27 | 36.19 | 63.37 |
| Multimodal | UrbanCLIP | 42.95 | 72.45 | 42.02 | 60.86 | 39.96 | 37.76 | 63.22 |
| Visual+Graph | GeoCLR | 39.76 | 69.99 | 39.00 | 58.40 | 37.48 | 35.79 | 61.61 |
| Visual+Graph | SceneParse | 43.16 | 72.80 | 41.52 | 60.97 | 39.41 | 33.44 | 63.18 |
| Visual+Graph | HybridSGN | 46.48 | 74.55 | 44.55 | 63.36 | 43.68 | 42.05 | 65.93 |
| Textual+Graph | KT-BERT | 43.03 | 71.49 | 40.64 | 60.69 | 40.46 | 33.77 | 63.49 |
| Multimodal+Graph | SemiGTX | 43.11 | 73.73 | 42.57 | 61.13 | 42.65 | 41.23 | 65.50 |
| Multimodal+Graph | USPM | 47.01 | 77.24 | 44.04 | 64.43 | 43.69 | 41.04 | 66.02 |
| Multimodal+Graph | **MetaStreet** | **51.42** | **77.52** | **48.34** | **66.94** | **45.52** | **43.46** | **67.44** |

*Table 2.* Performance comparison on the Xi'an dataset, with the best results in bold and the second-best underlined.

| Category | Method | House Price Prediction | | | Economic Activity Prediction | | |
|---|---|---|---|---|---|---|---|
| | | ACC | Macro-F1 | MRR | ACC | Macro-F1 | MRR |
| Visual | SemSeg | 31.74 | 31.49 | 57.52 | 33.87 | 33.34 | 59.02 |
| Visual | ResNet | 34.37 | 34.32 | 59.55 | 35.79 | 35.07 | 60.58 |
| Visual | READ | 35.17 | 35.04 | 59.81 | 36.03 | 35.69 | 60.65 |
| VLM | Qwen3-VL-235B | 29.03 | 22.61 | 55.81 | 36.46 | 36.23 | 61.84 |
| VLM | Qwen3-VL-8B (FT) | 43.62 | 43.72 | 65.43 | 40.87 | 41.15 | 64.49 |
| Multimodal | UrbanCLIP | 37.39 | 35.49 | 61.52 | 39.69 | 39.74 | 63.34 |
| Visual+Graph | GeoCLR | 40.73 | 39.78 | 64.15 | 38.32 | 38.09 | 62.35 |
| Visual+Graph | SceneParse | 46.19 | 46.57 | 67.63 | 41.24 | 41.05 | 64.37 |
| Visual+Graph | HybridSGN | 47.02 | 46.11 | 67.54 | 43.65 | 43.42 | 65.73 |
| Textual+Graph | KT-BERT | 44.53 | 39.15 | 66.77 | 42.89 | 42.06 | 65.97 |
| Multimodal+Graph | SemiGTX | 48.49 | 47.21 | 70.02 | 42.33 | 42.26 | 65.52 |
| Multimodal+Graph | USPM | 50.14 | 50.27 | 70.82 | 43.62 | 43.22 | 65.99 |
| Multimodal+Graph | **MetaStreet** | **53.26** | **52.64** | **72.19** | **45.77** | **45.49** | **67.01** |

**(T1) Street Function Prediction** (Wuhan): A 10-class classification task where streets are labeled with their primary functional type (residential, commercial, industrial, etc.) based on the EULUC-China land use dataset (Zhang et al., 2023). The training set includes 295 labeled streets (5.4% annotation rate). Since the EULUC-China dataset provides region-level rather than street-level annotations, we follow (Zhang et al., 2023) and assign each street the label of its nearest annotated region for ground-truth construction.

**(T2) Economic Activity Prediction** (Wuhan & Xi'an): A 4-class classification task using POI density as a proxy for economic vitality. Following (Li et al., 2022; Chen et al., 2024), POI counts are discretized into quartiles. We use 310 labeled streets (5.7%) for Wuhan and 270 (5.5%) for Xi'an.

**(T3) House Price Prediction** (Xi'an): A 4-class classification task based on average property prices from Anjuke[2], discretized into quartiles. We use 271 labeled streets (5.5%) for training.

### 4.1.3. BASELINES AND EVALUATION METRICS

We compare against 12 representative methods spanning five categories: visual-only methods (SemSeg (Zhou et al., 2019), ResNet (He et al., 2016), READ (Han et al., 2020)), vision-language model (VLM) baselines (Qwen3-VL-235B and Qwen3-VL-8B (FT) (Bai et al., 2025)), multimodal methods (UrbanCLIP (Yan et al., 2024)), graph-enhanced visual/textual methods (GeoCLR (Li et al., 2022), SceneParse (Lee et al., 2021), HybridSGN (Zhang et al., 2024), KT-BERT (Zhang et al., 2023)), and graph-enhanced multimodal methods (SemiGTX (Cao et al., 2025), USPM (Chen et al., 2024)). Detailed baseline descriptions are provided in Appendix B.

We adopt Accuracy (ACC), F1-score (Weighted-F1 for T1; Macro-F1 for T2/T3), and Mean Reciprocal Rank (MRR) as evaluation metrics. For the 10-class T1 task, we additionally report ACC@K (K $\in \{1, 3\}$). Detailed metric definitions are provided in Appendix C.

---

[2]https://www.anjuke.com

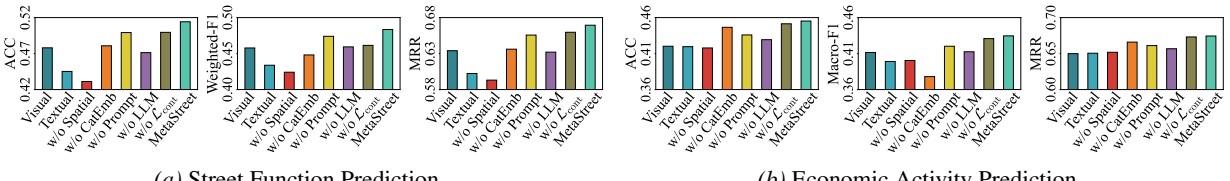

*(a)* Street Function Prediction        *(b)* Economic Activity Prediction

*Figure 4.* Ablation study results on Wuhan dataset.

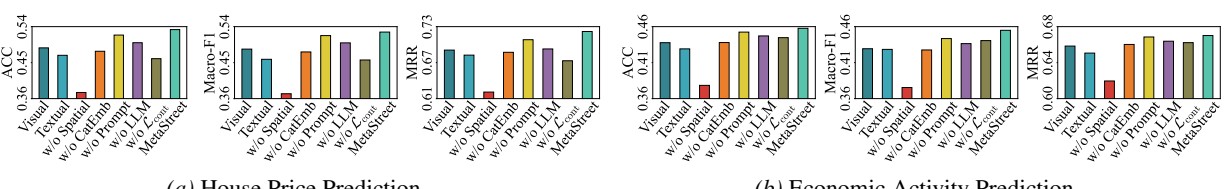

*(a)* House Price Prediction        *(b)* Economic Activity Prediction

*Figure 5.* Ablation study results on Xi'an dataset.

### 4.1.4. IMPLEMENTATION DETAILS

For the semantic-spatial visual encoder, we use a pretrained segmentation model (Zhou et al., 2019) with 150 categories, from which we select 48 urban-relevant categories by filtering out indoor and rare objects. Category embedding dimension $d_c = 16$; visual embedding dimension $d = 768$. For the textual encoder, we employ Qwen3-8B (Yang et al., 2025) with frozen weights. For graph construction, we set $k = 5$ neighbors. We use a 2-layer GAT with dropout rate 0.6. Negative sample size $N_{neg}$ is 64 for T1 and 8 for T2/T3. Loss weight $\lambda$ is tuned per task (0.5 for T1, 0.9 for T2 on Wuhan, 0.5 for T2 on Xi'an, and 0.1 for T3). We train using Adam optimizer with learning rate $5 \times 10^{-4}$. All results are averaged over 5 independent runs.

### 4.2. Overall Performance

Tables 1 and 2 present the comparison results. We make the following observations:

- Simple visual representations prove insufficient. SemSeg, despite extracting meaningful urban objects, performs worst due to its independent treatment of object categories. ResNet and READ, relying on generic visual features, also underperform, highlighting the need for urban-specific visual encoding that captures compositional structure.

- Graph-based spatial modeling is essential. Methods incorporating spatial graphs (GeoCLR, SceneParse, HybridSGN, KT-BERT) consistently outperform their non-graph counterparts. This confirms that geographic context and inter-street dependencies provide crucial complementary signals for street-level prediction.

- Multimodal approaches generally outperform unimodal ones, confirming that textual semantics complement vi-

sual features. Notably, VLM baselines (Qwen3-VL) underperform structured multimodal methods despite strong visual reasoning capabilities, indicating that street-level socioeconomic prediction requires explicit spatial modeling and task-aware semantic extraction beyond what end-to-end vision-language models currently offer.

- MetaStreet consistently outperforms all baselines. Across all three tasks and both cities, MetaStreet achieves the best performance on every metric. Compared with the best competing baseline in each task, MetaStreet improves ACC by 9.38% on street function prediction, 4.19% on economic activity prediction in Wuhan, 6.22% on house price prediction, and 4.86% on economic activity prediction in Xi'an. On average, MetaStreet achieves 6.16% relative improvement in ACC over the best competing baselines. A paired $t$-test confirms statistical significance ($p < 0.01$) for all improvements. Consistent gains are also observed on the NYC dataset (Appendix F).

### 4.3. Ablation Study

To quantify the contribution of each component, we evaluate the following variants: *Visual* (removes textual encoder), *Textual* (removes visual encoder), *w/o Spatial* (removes adjacency matrix $\mathbf{A}_{i,j}$), *w/o CatEmb* (removes category embeddings $\mathbf{C}$), *w/o Prompt* (removes task-specific prompts), *w/o LLM* (replaces Qwen with BERT), and *w/o $\mathcal{L}_{con}$* (removes contrastive loss). Detailed variant descriptions are provided in Appendix D. Results in Figures 4 and 5 reveal the following insights:

- Both modalities are essential. Removing either the visual encoder (*Textual*) or the textual encoder (*Visual*) leads to substantial performance degradation across all tasks, confirming that the two modalities capture complementary information: visual features encode physical scene

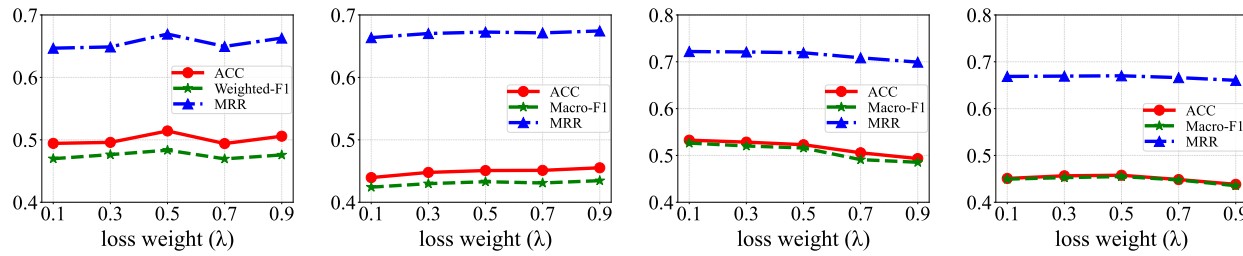

*(a)* Function Prediction (Wuhan)*(b)* Economic Prediction (Wuhan)*(c)* House Price Prediction (Xi'an)*(d)* Economic Prediction (Xi'an)

*Figure 6.* Impact of the loss weight $\lambda$ across different tasks and datasets.

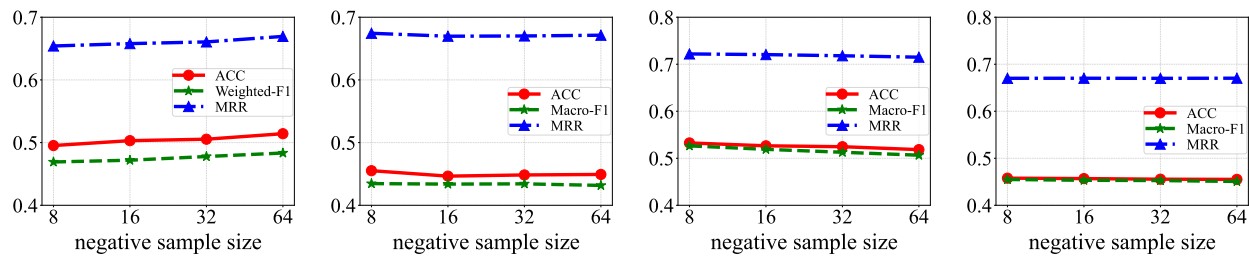

*(a)* Function Prediction (Wuhan)*(b)* Economic Prediction (Wuhan)*(c)* House Price Prediction (Xi'an)*(d)* Economic Prediction (Xi'an)

*Figure 7.* Impact of the number of negative samples across different tasks and datasets.

layouts while textual features provide high-level semantic abstraction.

- Semantic-spatial modeling is critical for visual encoding. The *w/o Spatial* variant shows notable performance drops, validating that spatial arrangements of urban objects carry discriminative information beyond category proportions. The *w/o CatEmb* variant exhibits one of the most severe degradations, demonstrating that learned category embeddings effectively encode urban-specific semantic relationships that raw features cannot capture.

- Task-aware prompting enhances textual encoding. Performance degrades in the *w/o Prompt* variant, indicating that task-specific prompts successfully guide the LLM to focus on prediction-relevant attributes rather than generic scene descriptions. The further drop in *w/o LLM* validates that large language models offer substantial advantages over smaller encoders like BERT for understanding complex street-level semantics.

- Contrastive learning improves semi-supervised learning. Removing $\mathcal{L}_{con}$ reduces performance across all metrics, confirming that our geography-aware contrastive learning effectively leverages unlabeled streets by extending contrastive supervision through spatial proximity.

### 4.4. Parameter Sensitivity and Efficiency

We analyze the sensitivity of MetaStreet to two key hyperparameters: the loss weight $\lambda$ that balances supervised

and contrastive losses, and the negative sample size $N_{neg}$ in contrastive learning. Results are shown in Figures 6 and 7.

For the loss weight $\lambda$, we find that optimal values vary by task characteristics. Street function prediction (10 classes) benefits from balanced supervision ($\lambda = 0.5$), while house price prediction favors stronger contrastive learning ($\lambda = 0.1$), likely due to strong spatial autocorrelation in property values. Economic activity prediction remains stable across all $\lambda$ values.

For negative sample size, tasks with more classes (T1: 10 classes) benefit from larger $N_{neg}$ up to 64, while tasks with fewer classes (T2, T3: 4 classes) remain stable across all settings, indicating that a smaller negative set suffices for coarser-grained classification.

We also evaluate computational efficiency by comparing training time per epoch. As shown in Appendix E, MetaStreet maintains competitive efficiency (16-19 seconds/epoch) comparable to USPM, while achieving substantial accuracy gains. This demonstrates a favorable accuracy-efficiency trade-off.

## 5. Related Work

### 5.1. Street View Image Representation

Street view imagery provides fine-grained, ground-level observations of urban environments (Li et al., 2022; Li & Zhou, 2025). Existing methods fall into two categories. The first

extracts visual features using pretrained encoders such as ResNet (He et al., 2016) or Vision Transformers (Dosovitskiy et al., 2021), often combined with spatially-guided contrastive learning (Li et al., 2022). However, these methods produce domain-agnostic features without explicit modeling of urban scene structure. The second parses images via semantic segmentation (Zhang et al., 2024; Zhou et al., 2019), but aggregates outputs as proportion vectors that ignore spatial layouts. Our semantic-spatial visual encoder addresses both limitations by jointly modeling category co-occurrence and spatial adjacency.

### 5.2. Multimodal Urban Computing

Recent studies integrate visual and textual modalities for urban understanding (Li et al., 2024b; Yan et al., 2024; Zhang et al., 2023; Li et al., 2026). UrbanCLIP (Yan et al., 2024) trains image encoders with LLM-generated captions via contrastive learning. KT-BERT (Zhang et al., 2023) encodes scene descriptions with BERT. However, these methods generate task-agnostic captions that may emphasize irrelevant attributes. Our task-aware textual encoder uses task-specific prompts to guide LLMs toward prediction-relevant features.

### 5.3. Semi-Supervised Learning for Urban Analysis

Street-level labels are scarce, motivating semi-supervised approaches. Graph-based methods (Chen et al., 2024; Cao et al., 2025) propagate information from labeled to unlabeled nodes via spatial adjacency. USPM (Chen et al., 2024) combines visual and textual features with graph propagation but treats unlabeled nodes passively. Contrastive learning (Chen et al., 2020; Khosla et al., 2020) offers an alternative, yet standard approaches either risk false negatives or limit participation to labeled samples. Our geography-aware graph contrastive learning addresses this by enabling unlabeled nodes to inherit contrastive supervision from labeled neighbors based on spatial autocorrelation.

## 6. Conclusion

We presented MetaStreet, a semi-supervised multimodal framework for street-level socioeconomic prediction. MetaStreet introduces a semantic-spatial visual encoder that jointly models object co-occurrence and spatial adjacency, a task-aware textual encoder that guides LLMs via task-specific prompts, and a geography-aware graph contrastive learning module that extends supervision to unlabeled streets through spatial autocorrelation. Experiments across two cities and three tasks validate the effectiveness of each component: semantic-spatial encoding outperforms pretrained encoders and proportion-based methods, task-specific prompts improve over generic captioning, and geography-aware contrastive learning provides consistent gains under label scarcity. Overall, MetaStreet achieves

6.16% relative improvement over the strongest baseline. Future work will explore cross-city transfer learning to improve generalization with minimal target-domain labels.

**Limitations.** MetaStreet builds on pretrained perception modules (semantic segmentation and image captioning), so errors from these upstream components can propagate downstream, particularly for fine-grained urban categories underrepresented in pretraining data. In addition, the current framework targets static urban analysis and does not model temporal evolution, which is important given that socioeconomic conditions shift over time. Incorporating uncertainty-aware perception and multi-temporal extensions are promising directions for future work.

## Acknowledgments

This work was supported in part by the National Natural Science Foundation of China under Grant No. 62572274, the Key Scientific and Technological Innovation Project of Shandong Province under Grant No. 2024CXGC010113 and 2024CXG010213, and the Open Fund of Key Laboratory of Urban Land Resources Monitoring and Simulation, Ministry of Natural Resources.

## Impact Statement

This paper advances multimodal semi-supervised learning with contributions applicable beyond urban analytics. The semantic-spatial visual encoding paradigm offers a principled approach to modeling compositional structure in scene understanding, applicable to other domains where object arrangements carry semantic meaning. The geography-aware graph contrastive learning framework demonstrates how domain-specific priors can extend supervision to unlabeled data, a strategy transferable to any setting where relational structure implies semantic similarity. For practitioners, MetaStreet enables scalable and fine-grained measurement of socioeconomic conditions, supporting evidence-based urban planning, equitable resource allocation, and data-driven policy-making in urban governance.

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

# A. Task-Specific Prompt Templates

We design task-specific prompts to guide the LLM toward extracting prediction-relevant features from street-level descriptions. Each prompt includes a role definition, specific objectives tailored to the prediction task, and the input data format. Below we provide the complete prompt templates along with example inputs.

## A.1. Street Function Prediction Prompt

You are a city function summarizer. Read several photo captions and the visual coverage ratio of the elements in the photos from a street in Wuhan, China, then write a continuous paragraph in English (250-400 words).

Objectives: 1. Describe *only* the elements related to the street function (residential, business office, commercial services, industry, transportation stations, administration, education, medical, sports and culture, parks and green space). 2. Combine similar elements in different captions (multiple "convenience stores" → "several convenience stores"). 3. Start the paragraph with the overall functional characteristics (primary and secondary uses). 4. Use quantity or intensity adjectives when obvious. 5. Omit anything that is not functionally relevant (weather, sky, shooting angle, people's clothing, lighting).

Input Data:

Image 1 Description: The street view shows a person riding a motorcycle down a tree-lined pathway. There are several trees on both sides of the path, providing shade and a pleasant atmosphere. The pathway is lined with white fences, indicating a well-maintained and organized area.

Proportions: Road (4%), Building (3%), Sky (7%), Sidewalk (49%), Wall (7%), Ceiling (24%), Floor (4%), Mountain (2%).

Image 2 Description: The street view shows a tree-lined street with a car parked on the side. There are no visible buildings or infrastructure, and no human activities are evident. The street appears to be a quiet, peaceful location with a natural setting.

Proportions: Road (1%), Sky (29%), Sidewalk (31%), Wall (19%), Ceiling (8%), Floor (9%), Bus (2%), Pole (1%).

...

## A.2. Economic Activity Prediction Prompt

You are a socioeconomic indicator analyst. Read several photo captions and the visual coverage ratio of the elements in the photos from a street in Wuhan, China, then write a continuous paragraph in English (250-400 words).

Objectives: 1. Describe *only* the elements that indicate socioeconomic conditions (number and density of POIs, commercial activity levels, infrastructure quality, building conditions, accessibility features, economic diversity). 2. Combine similar elements in different captions (multiple "convenience stores" → "several convenience stores indicate moderate commercial density"). 3. Start the paragraph with the overall socioeconomic characteristics (number and density of POIs, economic vitality, infrastructure development level, commercial density). 4. Use quantity or intensity adjectives to describe economic indicators when obvious (high/low density, well-maintained/deteriorated, diverse/limited). 5. Omit anything that is not socioeconomically relevant (weather, sky, shooting angle).

Input Data:

...

## A.3. House Price Prediction Prompt

You are a real estate environment analyst. Read several photo captions and the visual coverage ratio of the elements in the photos from a street in Xi'an, China, then write a continuous paragraph in English (250-400 words).

Objectives: 1. Describe *only* the elements that may influence property values (building quality and architectural style, neighborhood aesthetics, greenery coverage and environmental quality, proximity to amenities, street cleanliness and maintenance level). 2. Combine similar elements in different captions to provide an overall assessment. 3. Start the paragraph with the overall neighborhood quality characteristics. 4. Use descriptive adjectives to indicate property value implications (upscale/modest, well-maintained/neglected, desirable/undesirable). 5. Omit anything that does not affect property valuation (weather conditions, temporary objects, shooting angle).

Input Data:

...

# B. Baseline Descriptions

We compare against 12 representative methods spanning five categories:

**Visual-only methods:**

- **SemSeg** (Zhou et al., 2019): A pretrained semantic segmentation model that divides images into object categories, forming a representation vector from category proportions.
- **ResNet** (He et al., 2016): A widely used deep learning architecture pretrained on ImageNet for generating visual representations.
- **READ** (Han et al., 2020): A semi-supervised model utilizing teacher-student networks with pretrained models to learn image representations from partially labeled data.

**Multimodal methods:**

- **UrbanCLIP** (Yan et al., 2024): A vision-language framework that combines LLM-generated textual descriptions with contrastive learning to train an image encoder for urban imagery.

**VLM baselines:**

- **Qwen3-VL-235B** (Bai et al., 2025): A zero-shot multimodal large language model baseline that is prompted to directly predict street-level socioeconomic labels from street-view imagery.
- **Qwen3-VL-8B (FT)** (Bai et al., 2025): A fine-tuned multimodal large language model baseline that is trained to directly predict street-level socioeconomic labels from street-view imagery.

**Graph-enhanced visual/textual methods:**

- **GeoCLR** (Li et al., 2022): A geo-aware contrastive learning method that augments street view images and incorporates spatial neighbors to enhance visual representations.
- **SceneParse** (Lee et al., 2021): A method that extracts semantic segmentation features and constructs spatial graphs to capture relationships between images for socioeconomic prediction.
- **HybridSGN** (Zhang et al., 2024): A hybrid framework integrating semantic segmentation and graph neural networks with attention mechanisms to model visual and spatial relationships.
- **KT-BERT** (Zhang et al., 2023): A method that extracts urban entities and spatial correlations from street view images, encoding generated descriptions with BERT.

**Graph-enhanced multimodal methods:**

- **SemiGTX** (Cao et al., 2025): A semi-supervised graph framework that integrates multimodal urban data by balancing self-supervised and regression loss.
- **USPM** (Chen et al., 2024): A state-of-the-art semi-supervised street profiling model that combines street view imagery, spatial adjacency, and LLM-generated textual descriptions.

# C. Evaluation Metrics

We adopt the following evaluation metrics:

- **Accuracy (ACC)**: The proportion of correctly classified samples. For the 10-class street function prediction task (T1), we additionally report ACC@K, which measures whether the true label appears in the top-K predictions.

- **F1-score**: We use Weighted-F1 for T1 to account for class imbalance, and Macro-F1 for T2 and T3 where classes are more balanced (quartile-based discretization).

- **Mean Reciprocal Rank (MRR)**: Defined as $\mathrm{MRR} = \frac{1}{n} \sum_{i=1}^{n} \frac{1}{\mathrm{rank}_i}$, where $\mathrm{rank}_i$ is the rank of the true label in the predicted probability distribution. MRR captures ranking quality beyond top-1 accuracy.

*Table 3.* Training Time Comparison (seconds/epoch) on NVIDIA 2080 Ti GPU

| Method | Wuhan | | Xi'an | |
|---|---|---|---|---|
| | T1 | T2 | T2 | T3 |
| KT-BERT | 0.10 | 0.08 | 0.07 | 0.06 |
| USPM | 16.20 | 16.21 | 12.53 | 14.66 |
| HybridSGN | 21.94 | 21.84 | 13.24 | 14.99 |
| SemiGTX | 40.61 | 41.21 | 20.46 | 31.89 |
| MetaStreet | 17.49 | 18.79 | 16.72 | 17.34 |

## D. Ablation Study Variant Descriptions

We evaluate the following variants to quantify the contribution of each component:

- **Visual**: Removes the task-aware textual encoder and uses only the visual embedding $\mathbf{e}^{\text{vis}}$ as the street representation.

- **Textual**: Removes the semantic-spatial visual encoder and uses only the textual embedding $\mathbf{e}^{\text{txt}}$ as the street representation.

- **w/o Spatial**: Removes the spatial adjacency matrix $\mathbf{A}_{i,j}$ when computing visual embeddings, using only category embeddings weighted by proportions: $\mathbf{S}_{i,j} = \text{diag}(\mathbf{p}_{i,j}) \cdot \mathbf{C}$, which is then flattened into $\mathbf{v}_{i,j} \in \mathbb{R}^{K \cdot d_c}$.

- **w/o CatEmb**: Removes the learned category embedding matrix $\mathbf{C}$. The visual representation is computed as $\mathbf{v}_{i,j} = \tilde{\mathbf{A}}_{i,j} \cdot \mathbf{p}_{i,j} \in \mathbb{R}^K$, which aggregates category proportions through spatial adjacency without semantic embeddings.

- **w/o Prompt**: Removes the task-specific prompt $p_{\text{task}}$ from the textual encoder, feeding only the aggregated descriptive captions of street view images into the LLM.

- **w/o LLM**: Replaces the LLM (Qwen3-8B) with a pretrained BERT-base model in the textual encoder.

- **w/o $\mathcal{L}_{\text{con}}$**: Removes the contrastive learning loss $\mathcal{L}_{\text{con}}$ in the graph learning module and retains only the supervised cross-entropy loss $\mathcal{L}_{\text{sup}}$ for training.

## E. Computational Efficiency Analysis

Table 3 compares training time per epoch on an NVIDIA 2080 Ti GPU. We compare MetaStreet against representative graph-based methods that have comparable model complexity. MetaStreet demonstrates competitive computational efficiency, with training times ranging from 16.72s to 18.79s per epoch across all tasks. This is comparable to USPM (12.53s-16.21s) and substantially faster than SemiGTX (20.46s-41.21s). KT-BERT shows the fastest training time but relies on a smaller BERT encoder rather than an LLM, resulting in inferior prediction performance.

Despite incorporating both semantic-spatial visual encoding and task-aware textual encoding along with a contrastive learning mechanism, MetaStreet achieves substantial accuracy gains (up to 4.41 percentage points) with moderate computational overhead. This demonstrates a favorable accuracy-efficiency trade-off suitable for urban computing applications where prediction quality is paramount.

## F. Additional Dataset Evaluation

We evaluate MetaStreet on a New York City (NYC) dataset to assess generalization beyond the Chinese cities used in the main experiments. The dataset is built following the same protocol as in Section 4.1.1.

The NYC dataset consists of 4,538 street segments, with 63,178 street view images collected from Mapillary and Google Street View. We leverage the American Community Survey (ACS, 2020–2024 5-year estimates) at the census tract level to define two socioeconomic tasks: (i) *Rent Price Prediction* based on median gross rent, and (ii) *Education Level Prediction* based on the share of population aged 25 and over holding a bachelor's degree or higher. Both indicators are discretized into quartiles to form 4-class classification tasks. Each street is assigned the label of its nearest annotated census tract, and 5.5% of streets are sampled as training labels.

Table 4 shows that MetaStreet consistently outperforms all compared baselines on both tasks, confirming that the framework generalizes to a geographically and culturally distinct urban context.

*Table 4.* Results on the additional NYC dataset, with the best results in bold and the second-best underlined.

| Category | Method | Rent Price (NYC) | | | Education Level (NYC) | | |
|---|---|---|---|---|---|---|---|
| | | ACC | Macro-F1 | MRR | ACC | Macro-F1 | MRR |
| Visual | SemSeg | 31.33 | 28.75 | 55.98 | 32.76 | 32.12 | 57.62 |
| Visual | ResNet | 34.80 | 32.10 | 59.23 | 33.64 | 32.48 | 58.55 |
| Visual | READ | 33.90 | 31.76 | 58.59 | 35.15 | 34.47 | 59.34 |
| VLM | Qwen3-VL-235B | 30.93 | 30.91 | 56.31 | 31.98 | 32.28 | 56.84 |
| VLM | Qwen3-VL-8B (FT) | 41.65 | 40.39 | 62.85 | 46.05 | 45.90 | 66.62 |
| Multimodal | UrbanCLIP | 34.91 | 33.18 | 58.97 | 35.25 | 34.58 | 59.49 |
| Visual+Graph | GeoCLR | 48.10 | 45.51 | 65.67 | 49.93 | 49.33 | 67.25 |
| Visual+Graph | SceneParse | 50.01 | 47.76 | 68.86 | 51.22 | 50.66 | 70.23 |
| Visual+Graph | HybridSGN | 50.72 | 48.11 | 69.12 | 53.88 | 53.25 | 72.06 |
| Textual+Graph | KT-BERT | 41.49 | 40.81 | 63.61 | 43.65 | 42.85 | 65.37 |
| Multimodal+Graph | SemiGTX | 51.72 | 50.87 | 70.47 | 51.29 | 51.86 | 71.01 |
| Multimodal+Graph | USPM | 52.82 | 51.52 | 71.07 | 53.46 | 53.54 | 72.25 |
| Multimodal+Graph | **MetaStreet** | **54.80** | **53.92** | **73.05** | **56.33** | **55.37** | **73.73** |

## G. Spatial Autocorrelation Assessment

Our geography-aware negative sampling (Section 3.3.2) assumes that geographically proximate streets tend to share socioeconomic characteristics. We verify this assumption quantitatively using the $k=5$ nearest-neighbor graph adopted throughout the experiments.

**Nominal task.** For street function prediction (10 classes), 70.8% of graph edges connect streets assigned to the same functional class, which represents a $2.66\times$ enrichment over the random expectation, indicating pronounced spatial clustering of urban functions.

**Ordinal tasks.** For the three ordinal prediction settings (economic activity in Wuhan and Xi'an, and house price in Xi'an), we compute Moran's $I$ on the $k$-NN graph, where Moran's $I$ ranges from $-1$ (perfect dispersion) to $+1$ (perfect clustering), with values near 0 indicating spatial randomness. As shown in Table 5, all three settings exhibit strong positive autocorrelation ($I = 0.46$–$0.78$, all $p \leq 0.001$). House price prediction shows the highest autocorrelation ($I = 0.78$), consistent with the well-known spatial dependence of property values and explaining why this task benefits most from contrastive learning.

These results confirm that the locality assumption underlying our negative inheritance strategy is empirically well-supported across all datasets and task types.

*Table 5.* Spatial autocorrelation measured by Moran's $I$ for the ordinal socioeconomic prediction tasks.

| Task | Metric | Value |
|---|---|---|
| Economic Activity (Wuhan) | Moran's $I$ | 0.46 |
| Economic Activity (Xi'an) | Moran's $I$ | 0.67 |
| House Price (Xi'an) | Moran's $I$ | 0.78 |

