# OpenReview forum: "MetaStreet: Semi-Supervised Multimodal Learning for Street-Level Socioeconomic Prediction"
_ICML.cc/2026/Conference — ICML 2026 regular_

### Official Review · Reviewer_KL2N · 2026-03-04

**Soundness:** 2
**Presentation:** 3
**Significance:** 3
**Originality:** 3
**Overall Recommendation:** 4
**Confidence:** 3

**Summary:**

This paper proposes MetaStreet, a semi-supervised multimodal framework for street-level socioeconomic prediction from street view imagery. The authors identify three challenges. (C1) Inadequate modeling of compositional structure in visual encoding (object co-occurrence and spatial adjacency). (C2) Task-agnostic textual descriptions that do not align with prediction tasks. (C3) Passive use of unlabeled data in existing graph-based methods. MetaStreet addresses these with three components. (1) A semantic-spatial visual encoder that combines pixel-wise segmentation, category embeddings learned via Skip-gram over co-occurrence, and an object adjacency matrix (4-connected pixel adjacencies) aggregated with proportions and category embeddings, then LSTM aggregation over images per street. (2) A task-aware textual encoder that aggregates image captions and segmentation-derived statistics into street documents and prepends task-specific prompts to a frozen LLM (Qwen3-8B) to extract prediction-relevant features. (3) Geography-aware graph contrastive learning that defines positives as the mean of 1-hop spatial neighbors and enables unlabeled streets to participate as anchors by inheriting negative candidates from labeled neighbors, using spatial autocorrelation. Experiments on two Chinese cities (Wuhan, Xi’an) across three tasks (street function 10-class, economic activity 4-class, house price 4-class) show consistent, statistically significant improvements over 10 baselines. Ablations (e.g. removing spatial adjacency, category embeddings, prompts, LLM, or contrastive loss) and hyperparameter analyses (loss weight, negative sample size) support the contribution of each component.

**Compliance With Llm Reviewing Policy:**

Affirmed.

**Final Justification:**

I maintain my Weak Accept recommendation. The paper presents a technically solid and well motivated multimodal framework for street level socioeconomic prediction, with consistent gains over strong baselines and a contribution that is meaningful for urban computing even if the scope is somewhat specialized. The rebuttal adequately addressed my main concerns about class distribution, prompt sensitivity, graph sensitivity, and evaluation transparency, and I encourage the authors to incorporate these added analyses into the final version because they make the empirical case clearer and more convincing.

**Key Questions For Authors:**

1. Could you report the per-class label distribution (counts or proportions) for each task (street function, economic activity, house price) so that readers can assess whether accuracy is skewed by imbalance and interpret Weighted-F1 and macro metrics?

2. How were the 10 baselines trained and tuned (e.g. hyperparameters, optimization, number of runs)? Was there a common protocol or tuning budget to ensure a fair comparison with MetaStreet?

3. How sensitive are results to the wording or length of the task-specific prompts? Could you add an ablation that removes only the segmentation-derived textual statistics from the LLM input (keeping prompts and captions) to separate their effect from that of the prompts?

4. How was k = 5 chosen for the k-NN street graph, and how sensitive is performance to k? Have you considered or could you report results with alternative graph constructions (e.g. road-network topology)? Would you expect the approach to transfer to non-Chinese urban contexts, and under what conditions might the geography-aware negative sampling mislead (e.g. near land-use boundaries)?

**Limitations:**

Partially. The paper includes an Impact Statement (positive impact, transferability) but no dedicated Limitations subsection.

Suggested additions include dependence on segmentation and captioning quality, applicability beyond the two Chinese cities and current street view imagery coverage, lack of quantitative autocorrelation assessment and of failure-mode or fairness analysis, and unreported label distribution and baseline comparison protocol (which limit interpretability and comparability). Prompt sensitivity and graph design (e.g. k) are underexplored.

If the authors add a brief limitations subsection and optionally note deployment risks, the answer would be “yes.”

**Strengths And Weaknesses:**

**Soundness**
On the strengths side, the problem formulation (street network, SVI sets, semi-supervised prediction) and the three challenge and solution pairings are clear. The semantic-spatial visual encoder is well specified. Proportion-based composition, category co-occurrence via Skip-gram, and 4-connected spatial adjacency are clearly described and interpretable. The geography-aware negative sampling strategy is a concrete, justified way to activate unlabeled nodes. Task-aware prompting is operationalized with per-task prompts. Ablation studies show clear, interpretable drops when spatial adjacency, category embeddings, prompts, LLM, or contrastive loss are removed. Results are averaged over 5 independent runs with paired t-test (p < 0.01) for significance. Training-time comparison and hyperparameter sensitivity are reported.
On the weaknesses side, the following would strengthen the paper. The paper does not report the per-class label distribution for any task (street function, economic activity, house price). Without it, readers cannot assess whether reported accuracy is skewed by class imbalance or interpret the use of Weighted-F1 for T1. House price is formulated as 4-class classification (quartiles) but the choice of classification over regression is not justified. Implementation details describe only MetaStreet. How baselines are trained and tuned (hyperparameters, tuning effort, same protocol) is not stated, so the fairness of the comparison is unclear. The adjacency matrix is used as raw counts plus identity with no row or degree normalization discussed. Raw counts could bias representations by segment size or boundary length. The geography-aware negative inheritance relies on spatial autocorrelation but there is no quantitative assessment of autocorrelation strength. Prompt sensitivity is not analyzed, and the ablation that removes task-specific prompts does not isolate the effect of segmentation-derived textual statistics from that of the prompts. The graph uses k = 5 neighbors with no sensitivity analysis. Main results tables report means and significance but not confidence intervals or standard errors. Street function labels are assigned from the nearest annotated region (EULUC), which can introduce noise at boundaries. Acknowledgment or quantification would contextualize the setup. There is no analysis of failure modes under heavier label sparsity, model calibration, or fairness across subgroups.

**Presentation**
The three challenges (C1 to C3) map clearly to the three components. Figure 2 illustrates the full pipeline and Figure 3 explains the adjacency computation. The connection between urban priors (spatial autocorrelation, compositionality) and design choices is well articulated. Equations are concise and notation is consistent. Some symbol definitions are dense. A short summary table of main symbols and a brief summary of experimental takeaways would improve clarity.

**Significance**
Street-level socioeconomic prediction is relevant for urban planning and policy, and semi-supervised multimodal design is practically important given label scarcity. The idea of activating unlabeled nodes via geography-aware contrastive learning is transferable to other urban or geographic settings. The scope is specialized to street-view-based prediction and limited to two Chinese cities with no transfer or domain-shift evaluation. Impact would be strengthened by evidence of generalization and by clearer reporting of label distribution and baseline comparison protocol.

**Originality**
The joint treatment of compositional visual encoding, task-aware textual extraction, and label-efficient graph contrastive learning in one framework is novel for this domain. The semantic-spatial encoding (category co-occurrence plus spatial adjacency) and the geography-aware contrastive supervision with unlabeled nodes as anchors are clearly articulated contributions. Skip-gram for category embeddings and LSTM for street aggregation are established building blocks. The originality lies in their combination and in the geography-aware negative sampling design, rather than in new algorithmic or theoretical results.

---

> ### Author Rebuttal · Authors · 2026-03-31
>
> We thank the reviewer for the suggestions. Due to space limits, most additional tables/figures are provided at the anonymous link: https://anonymous.4open.science/r/rebuttal-5B3C/.
>
> **W1&Q1: Per-Class Label Distribution.**
> We use Weighted-F1 for street function prediction and Macro-F1 for economic activity and house price prediction, justified by their class distributions: street function is inherently imbalanced (residential: 46.6\%), while economic activity and house price are quartile-based and roughly balanced. Detailed distributions are in the link.
>
> **W2: Justification for Classification over Regression.**
> Accurate house price prediction depends on many micro-level factors (e.g., floor area ratio, building age) not observable from street-view images. Our goal is to infer macroscopic socioeconomic patterns from visual and geographic signals, for which quartile-based classification provides appropriate granularity. This follows prior urban computing work, GeoCLR (CIKM 2022) and USPM (KDD 2024).
>
> **W3&Q2: Implementation details.**
> We use official code for 8/10 baselines and faithfully re-implemented GeoCLR and SemiGTX, whose code is unavailable. We searched optimal hyperparameters per method. All methods share identical data splits and protocol; results averaged over 5 runs.
>
> **W4: Adjacency Matrix Normalization.**
> We compare row normalization, symmetric normalization, and raw counts (results at the link). Raw counts perform best, as they encode both spatial adjacency and the absolute scale of adjacency, which is discriminative; normalization discards this scale, retaining only relative proportions.
>
> **W5: Spatial Autocorrelation Assessment.**
> We quantified spatial autocorrelation on all tasks using our k=5 nearest-neighbor graph. For street function prediction (10 classes), 70.8\% of edges connect same-class streets, 2.66× random expectation. For economic activity and house price, Moran's I (ranging from -1 to 1, where values closer to 1 indicate stronger spatial clustering) shows strong positive autocorrelation (I = 0.46-0.78, all $p \leq 0.001$), quantitatively supporting our negative sampling strategy.
>
> |Task|Moran's I|
> |-|-:|
> |Economic Activity (Wuhan)|0.46|
> |Economic Activity (Xi'an)|0.67|
> |House Price (Xi'an)|0.78|
>
> **W6&Q3: Prompt Sensitivity.**
> We conduct two ablations: (1) paraphrasing the task-specific prompt while preserving semantics, and (2) removing segmentation proportions from the LLM input. Paraphrased prompts yield comparable performance (within ±1.0 F1 across all tasks), confirming prompt robustness. Removing segmentation statistics has modest effect on street function task but a larger drop on economic activity and house price tasks (e.g., Macro-F1: 50.55 vs. 52.64 on house price), where quantitative proportions complement captions. Full results at the link.
>
> **W7: Graph Neighbor Sensitivity.**
> We tested $k \in \{3,5,7,9\}$. Performance is best at $k=5$, stable at $k=7$, and worse at $k=3$ and $k=9$. Full results at the link.
>
> **W8:**
> Standard deviations over 5 runs are provided at the link; will be added to the revision.
>
> **W9: Label Noise at Boundaries.**
> We adopt the street function dataset from KT-BERT (ISPRS 2023). Training labels were manually curated by the original authors, selecting streets with unambiguous functions to ensure label quality. Test labels are assigned from the nearest EULUC region, which may add boundary noise. However, urban functional zones typically span large spatial extents, and most streets lie within a single functional zone rather than at boundaries, so the affected proportion is limited. We will discuss this in the revision.
>
> **W10: Label Sparsity and Per-Class Performance.**
> Experimental results are provided at the link. (1) Performance under heavier label sparsity (training sizes from 100 to 300): MetaStreet remains robust down to 200 samples and declines slightly at 100. (2) Per-class performance breakdown for tasks in Wuhan: MetaStreet shows relatively balanced performance across classes; minority classes are harder for all methods, but MetaStreet still outperforms USPM.
>
> **Q4:**
> - **Graph Construction.** We follow KT-BERT (ISPRS 2023), which found k-NN superior to Queen contiguity, as k-NN captures broader spatial context beyond directly intersecting streets. k-sensitivity analysis is given in W7.
> - **Generalizability.** Please see our response to Reviewer GA9J-W2 for new results on NYC (US).
> - **Robustness of Negative Sampling.** Near land-use boundaries, negative inheritance may introduce false negatives, but our random sampling ensures a small number of false negatives does not dominate the contrastive objective.
>
> **Limitation:**
> We will add a Limitations subsection in the revision. Most points are addressed above: segmentation quality (Reviewer qC7i-W4), generalizability (Reviewer GA9J-W2), autocorrelation assessment (W5), failure modes (W10), label distribution (W1), baseline protocol (W3), prompt sensitivity (W6), and graph design (W7).

---

> > ### Author Rebuttal · Reviewer_KL2N · 2026-04-02
> >
> > We appreciate the authors’ detailed rebuttal and thank them for addressing our questions clearly. The additional clarifications on class distribution, prompt sensitivity, and graph sensitivity resolved our main concerns. The response improves the paper’s transparency and strengthens our understanding of the evaluation setup. We will maintain our original recommendation.

---

> > > ### Author Response · Authors · 2026-04-02
> > >
> > > We truly appreciate your thorough review and are glad it helped improve our work. As all concerns have been addressed and the other reviewers have also raised their scores in light of the rebuttal, we kindly hope you might consider raising your score as well to support our work. Please let us know if there are any points that we can develop further.

---

### Official Review · Reviewer_GA9J · 2026-03-09

**Soundness:** 3
**Presentation:** 2
**Significance:** 1
**Originality:** 3
**Overall Recommendation:** 4
**Confidence:** 2

**Summary:**

The paper addresses the task of street-level socioeconomic prediction using a multi-stage pipeline involving a semantic-spatial visual encoder, as task-aware text encoder, and geography-aware graph contrastive learning. The paper evaluates on urban datasets from two major Chinese cities, on ~5k street segments with 50k-100k corresponding images. It evaluates on street-function prediction (10-way), economic activity prediction (4-way), and house-price prediction (4-way). The paper demonstrates superior results against baselines that are visual-only, multimodal, and graph-enhanced.

**Compliance With Llm Reviewing Policy:**

Affirmed.

**Final Justification:**

See the rebuttal acknowledgement below.

> I think the technical execution of this work is reasonable, although I still have reservations about its broader significance to the community. Nevertheless, my main concerns have been addressed and I have raised my score accordingly.

**Key Questions For Authors:**

- How does a vision-language model like Qwen3-VL or GPT-4o perform on this socioeconomic prediction task?
- Could there be a simpler variant of the proposed method that still maintains its core novelty (the geography aware graph contrastive learning) and competitive performance?

**Limitations:**

Yes, the limitations and potential negative societal impact of the work are adequately discussed.

**Strengths And Weaknesses:**

Strengths
- The paper proposes geography aware contrastive learning to overcome the issue of unlabeled data.
- The paper conducts a thorough ablation study of its subcomponents.

Weaknesses
- The proposed method has many steps, when the task could be addressed with simpler data-driven baselines (e.g., VLMs like Qwen3-VL or GPT-4o). Qwen3 is used as a subcomponent in Sec. 3.2.2, but it is unclear how well Qwen3-VL itself would do on this task, without the proposed scaffolding.
- The paper evaluates on two cities, making it unclear whether its findings are broadly significant.

---

> ### Author Rebuttal · Authors · 2026-03-31
>
> We sincerely thank the reviewer for the insightful questions, which allow us to better clarify the positioning of our work.
>
> **W1&Q1: Necessity of the Proposed Framework over VLM Baselines.**
> We would like to clarify that the problem studied in this paper is not a single-image classification task, but a socioeconomic prediction task at the street-level spatial granularity. A recent study CityLens[1] evaluated 17 large language-vision models (e.g., GPT-4o-mini) on urban socioeconomic prediction across 17 cities, and found that current VLMs struggle to associate visual content with socioeconomic indicators, with most models achieving weak predictive performance. Our findings are consistent with this conclusion. In our setting, street-level socioeconomic prediction faces three inherent challenges: (1) each street contains multiple images (up to hundreds) requiring aggregation, (2) streets exhibit spatial dependencies, and (3) only ~5\% of streets are labeled. VLMs, regardless of their visual reasoning capabilities, perform independent inference on individual (or few) images without modeling inter-street spatial structure or leveraging unlabeled data.
>
> We compare against three VLM baselines: zero-shot Qwen3-VL-235B, fine-tuned Qwen3-VL-8B, and Qwen3-VL-8B+Graph. The first two are prompted to directly predict street-level labels, while the third feeds VLM features into our graph contrastive learning module. Results are reported below.
>
> Even Qwen3-VL-235B underperforms the fine-tuned 8B variant, indicating that scaling VLM capacity alone cannot compensate for the lack of spatial modeling. Adding graph structure (Qwen3-VL-8B+Graph) improves over standalone VLMs, confirming the value of spatial modeling, but still falls substantially behind MetaStreet. This confirms that our semantic-spatial visual encoder and task-aware textual encoder provide structured information that VLM features cannot substitute.
>
> |Method|Category|Street Function|(Wuhan)|||Economic Activity|(Wuhan)||
> |-|-|:-:|:-:|:-:|:-:|:-:|:-:|:-:|
> |||ACC@1|ACC@3|Weighted-F1|MRR|ACC|Macro-F1|MRR|
> |Qwen3-VL-8B (FT)|MLLM|37.54|72.69|39.38|57.68|38.27|36.19|63.37|
> |Qwen3-VL-235B|MLLM|35.92|68.38|34.53|54.88|34.67|32.65|60.96|
> |Qwen3-VL-8B+Graph|MLLM+Graph|38.22|70.13|36.04|58.21|43.04|39.31|64.79|
> |**MetaStreet**||**51.42**|**77.52**|**48.34**|**66.94**|**45.52**|**43.46**|**67.44**|
>
> |Method|Category|House Price|(Xi'an)||Economic Activity|(Xi'an)||
> |-|-|:-:|:-:|:-:|:-:|:-:|:-:|
> |||ACC|Macro-F1|MRR|ACC|Macro-F1|MRR|
> |Qwen3-VL-8B (FT)|MLLM|43.62|43.72|65.43|40.87|41.15|64.49|
> |Qwen3-VL-235B|MLLM|29.03|22.61|55.81|36.46|36.23|61.84|
> |Qwen3-VL-8B+Graph|MLLM+Graph|41.81|40.11|65.06|42.12|37.02|65.49|
> |**MetaStreet**||**53.26**|**52.64**|**72.19**|**45.77**|**45.49**|**67.01**|
>
> [1] Tianhui Liu, Jie Feng, Hetian Pang, Xin Zhang, Tianjian Ouyang, Zhiyuan Zhang, and Yong Li*. Citylens: Benchmarking large language-vision models for urban socioeconomic sensing.
>
> **W2: Generalization to Additional Cities.**
> To further validate the generalizability of MetaStreet, we collected a new dataset from New York City (US) and conducted rent price prediction and education level prediction tasks. Results are reported below. MetaStreet consistently outperforms all baselines on both tasks, confirming its generalizability across cities and indicators.
>
> |Method|Rent Price|(NYC)||Education Level|(NYC)||
> |-|:-:|:-:|:-:|:-:|:-:|:-:|
> ||ACC|Macro-F1|MRR|ACC|Macro-F1|MRR|
> |HybridSGN|50.72|48.11|69.12|53.88|53.25|72.06|
> |KT-BERT|41.49|40.81|63.61|43.65|42.85|65.37|
> |SemiGTX|51.72|50.87|70.47|51.29|51.86|71.01|
> |USPM|52.82|51.52|71.07|53.46|53.54|72.25|
> |**MetaStreet**|**54.80**|**53.92**|**73.05**|**56.33**|**55.37**|**73.73**|
>
> **Q2: Indispensability of the Visual and Textual Encoders.**
> The semantic-spatial visual encoder and task-aware textual encoder are not auxiliary scaffolding but core components addressing the challenges identified in **our response to W1\&Q1 above**: the visual encoder explicitly captures the spatial structure of visual elements in street view images and aggregates multiple images within each street, while the textual encoder provides task-specific high-level semantics that complement the fine-grained visual features.
>
> Two lines of evidence confirm their indispensability: **(1)** Our ablation study in the main paper (Figures 4-5) shows that removing either modality (Visual or Textual variants) leads to significant performance degradation. **(2)** The Qwen3-VL+Graph experiment **(see Tables in  W1&Q1)** demonstrates that even powerful VLM features combined with graph contrastive learning still fall substantially behind MetaStreet, confirming that the two encoders provide structured information that VLM features cannot substitute.

---

> > ### Author Rebuttal · Reviewer_GA9J · 2026-03-31
> >
> > I think the technical execution of this work is reasonable, although I still have reservations about its broader significance to the community. Nevertheless, my main concerns have been addressed and I have raised my score accordingly.

---

> > > ### Author Response · Authors · 2026-04-01
> > >
> > > Thank you for your thoughtful follow-up and for taking the time to review our rebuttal. Your constructive comments have greatly helped us improve and further strengthen the manuscript. We sincerely appreciate the positive reassessment and are pleased that our additional experiments and clarifications have addressed your main concerns.

---

### Official Review · Reviewer_qC7i · 2026-03-11

**Soundness:** 2
**Presentation:** 3
**Significance:** 3
**Originality:** 2
**Overall Recommendation:** 4
**Confidence:** 2

**Summary:**

This paper introduces MetaStreet, a semi-supervised multimodal framework designed for street-level socioeconomic prediction (e.g., land-use, economic activity, and property values). The framework addresses existing limitations in visual feature extraction and label scarcity by proposing three integrated components: (1) a semantic-spatial visual encoder that jointly models object co-occurrence and spatial adjacency at the semantic category level; (2) a task-aware textual encoder that steers LLMs toward prediction-relevant features via task-specific prompts; and (3) a geography-aware graph contrastive learning module that leverages spatial autocorrelation to extend contrastive supervision to unlabeled streets, enabling them to actively participate in representation learning. Experiments conducted in Wuhan and Xi'an demonstrate that MetaStreet achieves state-of-the-art performance across multiple metrics compared to established baselines.

**Compliance With Llm Reviewing Policy:**

Affirmed.

**Final Justification:**

The responses have adequately addressed my previous concerns, particularly regarding the design rationale of the textual encoder. I will raise my rating.

I also encourage the authors to summarize the new experiments presented in the rebuttal and incorporate key components into the final manuscript.
This will significantly enhance the completeness and persuasiveness of the work.

**Key Questions For Authors:**

Please see the Weaknesses

**Limitations:**

The authors have not adequately discussed the limitations of the work. While they mention general domain challenges in the Impact Statement, they do not provide a specific discussion on the internal failure modes, or the sensitivity of the framework to input data quality (e.g., segmentation accuracy).

**Strengths And Weaknesses:**

Strengths:
- The framework consistently outperforms a wide range of baselines across different tasks and cities, showing robust performance in accuracy and F1-score.
- The manuscript provides a thorough analysis of hyper-parameters and ablation, which helps in understanding the necessity of different modules.

Weaknesses:
- The Introduction identifies three challenges, but does not sufficiently explain the current state of development in the field or why this specific three-component interaction is the most effective solution for the problem, can you provide a more detailed justification?
- The "Image -> Caption -> LLM" pipeline introduces an intermediate step that likely discards significant fine-grained visual information compared to using a direct Multimodal Large Language Model, why was the captioning-based textual encoder chosen over a direct MLLM approach?
- Line 365-367: The text appears to mischaracterize USPM as the strongest baseline in certain contexts, which is not consistently supported by the numerical rankings in Tables
- Since the semantic-spatial encoding relies on accurate masks, how much does the overall framework performance degrade if the segmentation mIoU drops? Have you considered quantifying this impact through a sensitivity analysis using different segmentation backbones?

---

> ### Author Rebuttal · Authors · 2026-03-31
>
> **W1: Necessity of the Three-Component Interaction.**
> The core problem is predicting street-level socioeconomic indicators from extremely limited labels (~5\% annotation rate). Our three components form a dependency chain rather than parallel modules. With so few labels, leveraging spatial autocorrelation to extend supervision to unlabeled streets is indispensable, which is the role of our geography-aware graph contrastive learning module. The visual and textual encoders provide discriminative multimodal embeddings; the effectiveness of graph contrastive learning (especially negative inheritance) depends directly on embedding quality. Conversely, high-quality embeddings alone cannot overcome extreme label scarcity without graph-based refinement.
>
> Two lines of evidence confirm this. First, the ablation study in the main paper (Figures 4-5) shows that removing any component leads to consistent degradation. Second, the VLM comparison (see our response to **Reviewer GA9J, W1\&Q1**) shows that Qwen3-VL and Qwen3-VL+Graph fall substantially behind MetaStreet (e.g., ACC 35.92 and 38.22 vs. 51.42 on street function prediction), confirming that our dual-encoder design cannot be substituted by a single model, even with graph learning.
>
> Regarding field context, early approaches extract visual features via pretrained encoders (ResNet, ViT) or segmentation but neglect spatial layout; recent work (USPM, SemiGTX) incorporates multimodal signals with semi-supervised graph learning, yet visual representations remain spatially uninformed, textual descriptions task-agnostic, and unlabeled nodes passive. We will trace this progression in the revised Introduction.
>
> **W2: Motivation of the Captioning-Based Textual Encoder.**
> The textual encoder is designed to provide high-level semantic abstraction (e.g., functional positioning, commercial vitality), rather than to re-encode visual details as the reviewer might expect. Fine-grained visual structure, including object co-occurrence and spatial adjacency, is already captured by our semantic-spatial visual encoder. If the textual encoder also processed raw images via an MLLM, both channels would operate on overlapping inputs (raw pixels), producing redundant rather than complementary representations. The captioning step converts information from visual to linguistic space, and this modality conversion is the key mechanism for complementary signals. The caption pipeline also naturally supports multi-image aggregation (some streets contain over a hundred images) and task-specific prompting, which are difficult in a single MLLM forward pass.
>
> As shown below, replacing the textual encoder with MLLM(Qwen3-VL-8B) yields only marginal gains over the visual-only variant and even degrades F1 on both Economic Activity tasks, while MetaStreet consistently outperforms it across all metrics.
>
> |Variant|Street Function|(Wuhan)||Economic Activity|(Wuhan)||
> |-|:-:|:-:|:-:|:-:|:-:|:-:|
> || ACC|F1|MRR|ACC|F1|MRR|
> |Visual only|47.80|45.78|63.40|42.02|41.15|64.99|
> |MLLM as Textual|48.67|46.31|64.23|43.13|36.93|65.53|
> |**MetaStreet**|**51.42**|**48.34**|**66.94**|**45.52**|**43.46**|**67.44**|
>
> |Variant|House Price|(Xi'an)||Economic Activity|(Xi'an)||
> |-|:-:|:-:|:-:|:-:|:-:|:-:|
> ||ACC|F1|MRR|ACC|F1|MRR|
> |Visual only|48.69|48.38|69.09|43.77|42.93|65.84|
> |MLLM as Textual|49.54|48.82|70.05|42.32|38.54|64.95|
> |**MetaStreet**|**53.26**|**52.64**|**72.19**|**45.77**|**45.49**|**67.01**|
>
> **W3: Baseline Ranking.**
> We thank the reviewer for the careful reading. The original statement is imprecise: while USPM ranks best on the majority of metrics, HybridSGN outperforms it on several. We will correct this in the revision.
>
> **W4: Robustness to Segmentation Quality.**
> We replace PSPNet (ResNet50) with a lighter PSPNet (MobileNetV2), both pretrained on ADE20K (150 categories), with mIoU 41.26 and 34.84 respectively. A 15.6\% relative mIoU drop leads to only 1.8-3.5 pp downstream degradation, and MetaStreet (MobileNetV2) still matches the top baseline USPM, demonstrating robustness. This robustness arises because our semantic-spatial encoding operates at the category level rather than pixel level, so localized misclassifications are diluted during aggregation.
>
> |Seg. Model|Street Function|(Wuhan)||Economic Activity|(Wuhan)||
> |-|:-:|:-:|:-:|:-:|:-:|:-:|
> ||ACC|F1|MRR|ACC|F1|MRR|
> |MetaStreet (MobileNetV2)|48.05|44.89|64.02|42.07|41.01|65.03|
> |**MetaStreet (ResNet50, ours)**|**51.42**|**48.34**|**66.94**|**45.52**|**43.46**|**67.44**|
>
> |Seg. Model|House Price|(Xi'an)||Economic Activity|(Xi'an)||
> |-|:-:|:-:|:-:|:-:|:-:|:-:|
> ||ACC|F1|MRR|ACC|F1|MRR|
> |MetaStreet (MobileNetV2)|50.47|50.24|70.39|43.03|43.14|65.22|
> |**MetaStreet (ResNet50, ours)**|**53.26**|**52.64**|**72.19**|**45.77**|**45.49**|**67.01**|
>
> **Limitation:**
> We acknowledge that MetaStreet relies on pretrained segmentation and captioning models, and degradation in either may affect downstream performance. We will add a Limitations subsection in the revision.

---

> > ### Author Rebuttal · Reviewer_qC7i · 2026-04-01
> >
> > I thank the authors for their detailed rebuttal.
> > The responses have adequately addressed my previous concerns, particularly regarding the design rationale of the textual encoder.
> > I will raise my rating.

---

> > > ### Author Response · Authors · 2026-04-01
> > >
> > > Thank you for your insightful suggestions and the updated evaluation. We are glad that our additional experiments and clarifications have addressed your concerns. Your detailed comments have greatly helped us improve the clarity and quality of the paper, and we will carefully incorporate them into the revision.

---

### Decision · Program_Chairs · 2026-04-30

**Decision:**

Accept (regular)

**Comment:**

After the discussion phase, all reviewers recommended acceptance (3x Weak Accept) noting that the paper is clearly presented, the method has novelty, and that the results are strong with extensive ablations. All reviewers indicated that the rebuttal addressed their concerns, e.g., by clarifying the motivation behind the textual encoder and clarifying technical details. As a result, the AC decided to accept the paper. Please take the reviewer feedback into account when preparing the camera-ready version.